# Azulene—A Bright Core for Sensing and Imaging

**DOI:** 10.3390/molecules26020353

**Published:** 2021-01-12

**Authors:** Lloyd C. Murfin, Simon E. Lewis

**Affiliations:** Department of Chemistry, University of Bath, Bath BA2 7AY, UK; sl288@bath.ac.uk

**Keywords:** fluorescence, azulene, sensor, dosimeter, bioimaging, chemosensor, chemodosimeter

## Abstract

Azulene is a hydrocarbon isomer of naphthalene known for its unusual colour and fluorescence properties. Through the harnessing of these properties, the literature has been enriched with a series of chemical sensors and dosimeters with distinct colorimetric and fluorescence responses. This review focuses specifically on the latter of these phenomena. The review is subdivided into two sections. Section one discusses turn-on fluorescent sensors employing azulene, for which the literature is dominated by examples of the unusual phenomenon of azulene protonation-dependent fluorescence. Section two focuses on fluorescent azulenes that have been used in the context of biological sensing and imaging. To aid the reader, the azulene skeleton is highlighted in blue in each compound.

## 1. Introduction

Azulene, **1**, is an isomer of naphthalene, **2**, composed of fused 5- and 7-membered ring systems (Figure 1) and named for its vibrant blue colour. Unlike naphthalene, azulene is a non-alternant hydrocarbon, possessing nodal points at C-2 and C-6 of the HOMO and C-1 and C-3 of the LUMO [1]. The location of these nodes results in low electronic repulsion in the S_1_ singlet excited state, affording a relatively small HOMO-LUMO gap. Hence, the S_0_→S_1_ transition arises from absorption in the visible region. Conversely, in naphthalene, coefficient magnitudes remain consistent for each position in both the HOMO and LUMO, affording a greater electronic repulsion; the greater S_0_–S_1_ energy gap sees naphthalene absorb in the UV region. In a similar relationship, there is little change in the coefficient magnitudes of the HOMO and LUMO+1 of azulene, creating a large S_0_–S_2_ gap (and hence also a large S_1_–S_2_ gap). This produces a system where the transition S_2_→S_0_ is the predominant mode of fluorescence [2,3], violating Kasha’s rule [4].

By varying the nature of the functional group attached to the azulene core at positions where electron density is greatest, the wavelength of light absorbed can be fine-tuned [5]. Electron-withdrawing groups present at the C-1 and C-3 positions stabilise the HOMO and LUMO+1 (whilst having no effect on the LUMO), increasing the S_0_–S_1_ gap. The contrary argument is true for electron-donating groups at these positions: the HOMO and LUMO+1 are destabilised, reducing the S_0_–S_1_ gap [6]. S_2_ emission predominates when the S_2_–S_1_ gap is >10,000 cm^−1^ (cf. 14,000 cm^−1^ in azulene **1**), whilst a mixture of S_2_ and S_1_ fluorescence emission is present when the S_2_–S_1_ gap is 9000 cm^−1^–10,000 cm^−1^ [7]. When the gap is smaller than 9000 cm^−1^, the rate of internal conversion between S_2_ and S_1_ is greater than the rate of relaxation from S_2_, so fluorescence occurs from S_1_ [8].

Whilst the azulene core has been incorporated into a wide variety of colorimetric chemical probes (exploiting the vivid colours it is famous for) [9,10,11,12,13,14,15,16,17,18,19,20,21,22,23,24,25,26,27,28,29,30], this review will focus on those published probes that exploit the *fluorescent* nature of azulene. The nature of azulene fluorescence is highly dependent on substitution around the azulene core, which affects whether emission is more dominant from the S_2_ or S_1_ singlet excited state (a notable exception from this can be found in the work of Zhu et al., who showed that emission of azulene aldehydes can occur from the S_3_ or S_2_ excited state dependent on the nature of the hydrogen bonding present [31,32]).

A variety of fluorescent probes harnessing the azulene core have been reported. Such probes are traditionally composed of three parts: a recognition moiety (that interacts selectively with the desired analyte), a reporter moiety (that gives a measurable response upon analyte interaction—in this instance the azulene core) and a linker that connects the two moieties. If the probe undergoes an irreversible, usually covalent, interaction with the analyte, the probe is described as a chemodosimeter [33]. Reaction of the recognition group and the analyte usually results in a cleavage of the linker and release of the response moiety as a separate species. Chemodosimeters are highly selective due to their covalent interaction with an analyte. In contrast, chemosensors interact reversibly and non-covalently with an analyte. The inherent reversibility of chemosensors often leads to reduced selectivity, but allows for a real-time response to the analyte-binding event and can therefore be used to monitor changes in analyte concentration over time [34].

Fluorescent probes offer a non-invasive means of analyte detection. They are required in scenarios where a colorimetric response is unsuitable, i.e., in an environment that is already highly coloured, such as a biological matrix [35]. Upon analyte interaction, the reporter of a fluorescence probe will respond in one of three ways:Begin fluorescing—a “turn-on” response.Stop fluorescing—a “turn-off” response.Alter emission wavelength—a ratiometric probe, whereby response is recorded as a ratio of the emission of the product over the emission of starting material.

When considering probes for biological applications, probes that excite or emit in the red to near-infrared (NIR) window (650–900 nm) offer significant advantages, namely non-invasive deep-tissue penetration (µm to cm) and reduced background absorption from water and haemoglobin [36], whilst reducing the risk of sample damage and photo-bleaching common with higher energy UV light sources [37]. Autofluorescence from endogenous species should also be avoided, as such species typically excite or emit below 600 nm [38].

## 2. Discussion

### 2.1. Turn-On Azulene Fluorescent Sensors

The most common fluorescence sensing motif within the literature is that of azulene proton probes. By considering azulene as a cyclopentadienyl anion fused to a tropylium cation, it can be appreciated that electron density is increased at the 1- and 3-positions. Protonation at either of these positions is known to lead to a significant increase in fluorescence (Scheme 1, in which the nomenclature [1 + H]^+^ denotes an azulene protonated on the 5-membred ring) [39].

When considering synthetic access to chemical probes, the electronic nature of azulene, as shown in Scheme 1, is often exploited. The nucleophilicity of the 5-membered ring is known to facilitate the addition of electrophiles to the azulene core at the 1- and/or 3-positions (e.g., halides [40], sulfonium salts [41]). Functionality can also be introduced to the azulene core by preferential C-H borylation of the 2-position [42], amination of the 6-position by vicarious nucleophilic substitution [43] or through the ground-up synthesis of the azulene skeleton. For example, the work by Nozoe et al. details how substituted azulenes can easily be synthesised in high yield from troponoids [44]. The following selection will highlight a range of azulene motifs that have specifically exploited the increased nucleophilicity of the azulene 1- and 3-positions to protonate the azulene core and generate turn-on fluorescence species.

Protonation-dependence fluorescence of azulene has been reported multiple times in the azulene-oligomer field [45]. In 2009, Xu et al. synthesised a series of 1,3-difluorenyl substituted azulene monomer (**M1**, **M2**) and polymer (**P5**, **P6**) macromolecules (Figure 2) that were shown to exhibit turn-on fluorescence upon titration with trifluoracetic acid (TFA) [46]. It was found that, upon protonation, the LUMO–LUMO+1 gap of the monomers shrinks, affording a decrease in the S_1_–S_2_ gap. The authors attributed the resulting change in the HOMO–LUMO gap as the cause of fluorescence.

Similarly, Hawker et al. reported a ten-fold increase in fluorescence intensity of their azulene-thiophene oligomer **3a** upon treatment with TFA (Figure 3) [47]. Upon protonation of the azulene core, both the wavelengths of maximum absorbance and maximum fluorescence emission underwent a bathochromic shift. Subsequent efforts of Hawker and co-workers revealed that the protonation-based fluorescence of azulene was dependent on the nature, position and connectivity of the aromatic groups attached to the azulene core [48]. Oligomers **3b**–**3f** remained non-fluorescent upon treatment with TFA. However, (bis)furan **3g** was found to exhibit a fluorescent turn-on response when protonated.

In 2012, Venkatesan et al. reported the synthesis of a family of 2-alkynyl azulenes (**4a**–**4k**, Figure 4), showing that the fluorescence emission could be altered depending on the protonation state of the azulene [49]. In particular, pentafluorophenyl derivative **4d** (the most fluorescent species synthesised in the study) was seen to undergo a significant red-shift in emission when treated with TFA. Neutral **4d** was found to absorb strongly in the visible region at 310 and 394 nm, which the authors calculated (Time-Dependent Density Functional Theory at the PBE1PBE/6-31+G9(d) level) corresponds to the S_0_→S_4_ and S_0_→S_2_ transitions respectively (notably, the theoretical S_0_→S_1_ transition was experimentally absent). When treated with MeSO_3_H, a single, broad, red-shifted absorbance was observed for [4d + H]^+^ at 440 nm (corresponding to the S_0_→S_1_ transition). Neutral species **4d** was found to emit with low fluorescence intensity (λ_ex_ = 394 nm, λ_em_ = 424 nm), which doubled when treated with TBAF, and increased 12-fold when treated with *^n^*Pr_4_NBF_4_. Protonated species [4d + H]^+^ was found to fluoresce ~250x as intensely as the neutral species, with a broad red-shifted emission at 487 nm. The red-shift in absorbance and emission upon protonation was observed with all species tested, with fluorescence emissions occurring from 443 nm all the way to 750 nm.

The following year, Venkatesan et al. published a follow-up article on the stimuli-responsive behaviour of di(phenylethynyl) azulenes species **5a**–**5d** (Figure 5) [50]. All neutral species showed three distinct absorbance regions: >500 nm (S_0_→S_1_), 500–370 nm (S_0_→S_2_) and <370 nm (S_0_→S_3_), with the most intense absorptions occurring at 409 (**5d**), 418 (**5b**), 423 (**5a**) and 446 nm (**5d**). The fluorescence emissions of **5a** (465 nm) and **5b** (432 nm) were found to be very weak, with intensity increasing slightly for **5c** (484 nm). Conversely, **5d** gave a strong, sharp emission at 416 nm. TF-DFT calculations suggested S_2_→S_0_ as the dominant mode of fluorescence in all cases. Upon protonation all species exhibited the previously observed red-shift in absorbances, with maxima at 703, 689, 525 and 490 nm for [5a + H]^+^ to [5b + H]^+^, respectively. Absorbance intensities of [5c + H]^+^ and [5d + H]^+^ were significantly greater than the other probes, which exhibited relatively low extinction coefficients. Fluorescent emissions of the protonated species also underwent a bathochromic shift relative to their neutral counterparts. Probes [5a + H]^+^ (722 nm) and [5b + H]^+^ (733 nm) remained relatively weakly fluorescent, whilst [5c + H]^+^ (571 nm) and [5d + H]^+^ (537 nm) were strongly fluorescent with the former now emitting with the greatest intensity. The authors commented that the S_2_→S_0_ emission was dominant for [5a + H]^+^, whilst S_1_→S_0_ emissions were dominant for the other three probes.

In 2015, Belfield et al. showed that 4-styrylguaiazulene derivates (**6a**–**6l**, Figure 6) were also able to show NIR fluorescent emission upon protonation of the guaiazulene core [51]. Similar to the work published by Venkatesan et al. in 2012, the nature of the absorption and emission of the guaiazulene derivatives was tuned by altering the nature of the attached conjugated aromatic system. Neutral species **6a**–**6l** were found to display weak S_1_→S_0_ fluorescent emission in DCM. Protonation of all species afforded a fluorescence turn-on response, with accompanying red-shift of emission. Significant Stokes shifts were observed, ranging from 67–182 nm. Fluorescence emission (S_1_→S_0_) of the compounds all occurred at relatively long wavelengths (512–768 nm), with compounds [6d + H]^+^–[6f + H]^+^ and [6j + H]^+^–[6l + H]^+^ emitting in the NIR region (650–950 nm). The authors proposed that upon protonation of the guaiazulene core, an intramolecular charge transfer (ICT) state is generated. Computational studies have been performed by Belfield et al. on analogous fluorescent guaiazulene species. The group were able to refine their computational model to obtain predicted fluorescence spectra which closely matched those obtained experimentally [52].

In 2018, Gao et al. explored the effect on fluorescence of substituting azulene at either the 2- or 6-position with aromatic moieties (**7a**–**f**, Figure 7) [53]. Similar to the previous observations, fluorescence was dependent on the protonation state of the azulene core. In the neutral state only **7d** was found to be strongly fluorescent (λ_em_ = 386 nm), whilst the remaining compounds emitted with weak intensities (λ_em_ = 408–447 nm). Upon protonation, red-shifted emissions with strong intensities were observed for **7a** (λ_em_ = 451 nm), **7b** (λ_em_ = 503 nm) and **7f** (λ_em_ = 500 nm). The remaining compounds were found to be relatively non-fluorescent in comparison.

Within the azulene protonation-dependent fluorescence field, there are multiple published examples reported by Shoji and co-workers [54]. A series of 2-arylazulenes were synthesised, of varying electronic nature, on the attached aryl moieties (Figure 8). Azulenes **8a**–**8c** and **9a**–**9h** were all found to undergo fluorescence enhancement when in acidic solution. The presence of the isopropyl group within **8b** and **8c** gave minor increases in quantum yield and decreases in Stokes shift. The fluorescence properties were found to be dependent on the electronic nature of group at the *para*-position of the phenyl substituents. Electron-donating moieties (**9a**, **9b**, **9e**) afforded lower quantum yields, fluorescence emission at longer wavelengths and greater Stokes shifts. Conversely, the presence of electron-withdrawing moieties (**9c**) resulted in greater quantum yields and emission at shorter wavelengths.

Shoji and co-workers have also shown that fused-azulene ring systems (Figure 9) can undergo protonation-mediated fluorescence. Fused azulenephthalimide **10** was shown to undergo a “turn-on” fluorescence response in acidic medium (λ_ex_ = 390 nm, λ_em_ = 444 nm) [55]. The group attributed this fluorescence enhancement to the formation of the tropylium cation—the potential quenching role of the azulene was inhibited, allowing for emission from the phthalimide core. Interestingly, the fluorescent properties of non-fused azulenephthalimides were shown to undergo only a small fluorescent enhancement in acidic solution [56]. A series of non-fused azulenephthalimides were synthesised, varying the nature of the substitution on the 5- and 7-membered rings of azulene. Only compounds bearing alkyl chains on the azulene 5-membered ring were shown to undergo a significant turn-on fluorescence response upon protonation (e.g., **11**, whereby the *amine of the phthalimide moiety becomes protonated*, rather than the azulene core). The cause of this fluorescence behaviour is unknown, but it has been suggested that it is caused by perturbing the nature of the donor–acceptor interaction between the azulene and phthalimide moieties.

Unlike the fused azulenephthalimide counterpart, fused azulenethiophene **12** showed moderately weak fluorescence emission in acidic solvent [57]. It was suggested that a non-radiative decay pathway was present due to the rotational freedom of the phenyl moiety bound to the thiophene. The fluorescent properties of analogues of **10**, **11** and **12** are reported in the Electronic Supplementary Information (ESI) of their corresponding publications, which show the fluorescence trends described above.

The most recent entry in the literature on protonation-dependent fluorescence of azulene was reported by Lewis, Kann and co-workers in 2020 [29]. They found that 1,3-difunctionalised azulenes **13a**–**e** (Figure 10) protonated upon the addition of TFA, for which a strong turn-on fluorescence response was reported in each instance. Of all the compounds assayed, azulene **13c** was found to undergo the greatest fluorescence enhancement when protonated (λ_ex_ = 266 nm, λ_em_ = 336 nm), for which the tricarbonyliron(diene) moiety did not lead to emission quenching.

Aside from protonation of the azulene core, azulene monoaldehydes have likewise been shown to exhibit alterations in their fluorescence when protonated (whereby protonation occurs on the aldehyde moiety). Aldehydes **14** and **15** (Figure 11), reported by Li, Gao and co-workers in 2008, were shown to give considerable redshifts (>100 nm) in their emissions when the solvent was changed from EtOH (**14** λ_em_ = 420 nm, **15** λ_em_ = 428 nm) to HClO_4_ (**14** λ_em_ = 552 nm, **15** λ_em_ = 530 nm) or TFA (**14** λ_em_ = 537 nm, **15** λ_em_ = 530 nm) [58]. The change in emission of dialdehyde **15** was similar enough to monoaldehyde **14** to suggest that only monoprotonation occurs in the former species.

Similarly, the 1,3-connected azulene imine ligands described by Jamali, Bagherzadeh and co-workers (Figure 12) showed considerable changes in their fluorescence properties upon protonation [59]. Both monoimine **16** and diimine **17** were found to be weakly emissive in CH_2_Cl_2_ (≈ λ_em_ = 450 nm for both species). Upon addition of TFA (10%), large redshifts in emission afforded strong fluorescence at λ_em_ = 610 nm and λ_em_ = 570 nm for monoimine **16** and diimine **17**, respectively. For both species, ^1^H-NMR studies confirmed that protonation occurs on the imine nitrogen and that, unlike dialdehyde **15**, diimine **17** undergoes protonation at both nitrogen atoms.

Fluorescence sensing for specific analytes, outside of azulene protonation, is scarce. In 2005, Eichen et al. published a “turn-on” fluorescent azulene sensor for fluoride [60]. The sensor, 1,3-di(2-pyrrolyl)azulene **18**, was found to be selective towards tetrabutylammonium (TBA) fluoride over other TBAX salts in both DMSO and DCM (where X = Cl**^−^**, Br^−^, I**^−^**, *p*-toluenesulfonate, BF_4_**^−^**, PF_6_**^−^**). Upon stoichiometric binding of fluoride, a ten-fold increase in fluorescence was observed (λ_ex_ = 355 nm, λ_em_ = 450 nm). The authors noted that the proposed S_2_→S_0_ fluorescence resulted from a change in conformation of the pyrrole rings. In the unbound species, the rings are expected to be in the plane of the azulene ring. Mechanistically, the authors proposed that fluoride was hydrogen-bonded to the pyrrole moieties, which in turn rotated out of the plane of the azulene (Scheme 2). With the pyrrole rings out of plane, the excited state of the complex was altered, allowing greater emission.

### 2.2. Fluorescent Azulenes in Biological Contexts

A range of azulenes have been employed in biological settings, including as anti-cancer agents [61,62,63], antiretrovirals [64], anti-inflammatories [65,66], erectile dysfunction treatments [67,68], anti-ulcer agents [69,70,71,72] and as a potential type II diabetes treatment [73]. The work by Pham et al. highlighted how an ^18^F-labelled azulene derivative can be used to image cancer cells [74]. By using COX2 as a biomarker, they were able to gain selective retention of the probe within breast cancer tumours in mice and image them via positron emission tomography. However, few biologically relevant azulenes have been tested for their fluorescence properties.

The most prominent entry in the field of azulene fluorescence in biological settings is the use of the non-canonical amino acid β-(1-azulenyl)-l-alanine (referred to in the literature as **Aal** and **AzAla** interchangeably), commonly used as a bioisostere of tryptophan (Figure 13). The first enantiomerically pure synthesis of **Aal** was published in 1999 by Moroder et al. through a key enantioselective deacetylation step using the enzyme acylase I [75]. It was found that **Aal** could be excited at both 276 and 339 nm to emit at 381 nm. The longer excitation wavelength was noted to be particularly useful, allowing selective excitation over tryptophan (λ_ex_ = 280 nm, λ_em_ = 381 nm).

In 2004, Pispisa et al. assessed the fluorescent properties of **Aal** in hexapeptides when placed at different distances from the fluorescent quencher 2,2′,6,6′-tetramethyl-1-oxyl-4-amino-4-piperidine carboxylic acid (TOAC) [76]. Two unique hexapeptides were synthesised, wherein the **Aal** group was either 1- or 2-amino acid residues away from TOAC in the primary structure (**A1T3** and **A1T4** respectively, Figure 14a). It was found that quenching of the **Aal** fluorescence was more significant when the TOAC was further away in the primary structure, due to the greater proximity of TOAC and **Aal** in the secondary structure affording a higher level of through-space quenching. The following year, Pispisa et al. went on to report the use of **Aal** as a fluorescent biomarker by incorporating the amino acid into a synthetic analogue of the antibiotic peptide trichogin GA IV [77]. The fluorescent peptide analogue, **A3** (Figure 14b), was used to probe the membrane activity of trichogin **GA IV**, for which the ability to selectively excite **Aal** over Trp was exploited.

**Aal** has also been used as a fluorescence quencher. In 2006, Koh and Lee incorporated fluorescein (as a fluorescence donor, λ_ex_ = 467 nm, λ_em_ = 506–560 nm) and **Aal** (as a fluorescence acceptor) into a synthetic oligomer mimic of naturally occurring β-amyloid (Aβ) peptides [78]. Fluorescence decay was monitored as function of increased peptide concentration and subsequent aggregation, for which the increase in the extent of conformational change increased the efficiency of the Förster resonance energy transfer (FRET) quenching mechanism.

**Aal** was shown to be a useful fluorophore for monitoring protein–-protein interactions by Korendovych et al. in 2013 [79]. Unlike Trp, **Aal** was shown to be insensitive to local environment changes (e.g., solvent exposure, local interactions) and therefore proposed to be more sensitive to weaker intrinsic quenchers (e.g., methionine). Due to the environmental sensitivity of Trp, the deconvolution of the effects of environment changes and quenching on fluorescence is difficult. To test this proposal, **Aal** was incorporated into the binding partner of the eukaryotic protein calmodulin (CaM), a 20-residue peptide of smooth muscle myosin light chain kinase (smMLCK). The ability to selectively excite **Aal** over Trp at 342 nm proved advantageous for monitoring the resultant protein–protein interaction for determining binding stoichiometry, without the interference of the environmentally sensitive Trp residue. Furthermore, the exchange of **Aal** for Trp within smMLCK was found to not significantly alter the smMLCK–CaM interaction.

Korendovych et al. further explored the effects of exchanging Trp for **Aal** in 2015, assessing if the presence of **Aal** would affect native ion channels [80]. Using the transmembrane domain of the M2 proton channel (M2TM) of the influenza A virus as their model, a key Trp residue was replaced with **Aal** (termed M2TM Trp41AzAla). Notably, the ion channel assembles into a tetramer of M2TM. In M2TM, the channel positions the Trp residue one helical turn below a key His residue responsible for proton conductance. Upon forming a 3:1, M2TM: M2TM Trp41AzAla tetramer, the protonation state of the His residue could be determined as a function of fluorescence of the **Aal** moiety (λ_ex_ = 342 nm, λ_em_ = 380 nm), whereby fluorescence intensity decreased when pH > 7 and increased when pH < 7.

Another instance in which **Aal** has been used in place of Trp was highlighted by the work by Kalesse et al. in 2018 [81]. The work details the total synthesis of **Argyrin C**, a biologically relevant cyclic octapeptide and the subsequent synthesis of **Aal argyrin C** in which a Trp residue (highlighted in red) was replaced with **Aal** (Figure 15). When incorporated into the octapeptide, the fluorescence intensity (λ_ex_ = 342 nm, λ_em_ = 380 nm) of **Aal** was reduced (~50% of the free Boc-**Aal**-OH residue) but was much greater than that of Trp, which was almost non-fluorescent at the same excitation wavelength. However, the authors noted that the circular dichroism (CD) spectrum of the altered peptide differs dramatically to that of the wild type. The work notably reported the synthesis of Boc-**Aal**-OH with a 93% yield over two steps.

Most recently, with regard to **Aal**, in 2019 Arnold et al. showed that by using an engineered tryptophan synthase, **Aal** could be synthesised in the near gram-scale (965 mg) in a single step from serine and azulene [82].

Azulenyl squaraines have found use in biological settings as NIR FRET quenchers for use in peptides. In 2002 Tung et al. developed **NIRQ_750_** (Figure 16), proposing that the extended π-system of the molecule would allow for long wavelength absorbance in the region of 650–900 nm [83]. It was found that **NIRQ_750_** absorbed broadly in the NIR region of 700–800 nm. Absorbance was found to vary with solvent polarity, whereby more polar solvents afforded a bathochromic shift in absorbance. To assess the quenching feasibility of **NIRQ_750_**, a fluorescent capase-3 peptide substrate was designed to incorporate **NIRQ_750_** at the N-terminus. At the opposite end of the peptide, attached to a cysteine residue, Alexa-680 C_2_ maleimide (λ_ex_ = 679 nm, λ_em_ = 702 nm) was used as a NIR-fluorescence donor. When tested against the capase-3 enzyme, a fluorescence response was reported. Tung et al. further contributed to the field the following year with the addition of **NIRQ_700_** (Figure 16) [84]. They proposed that a shorter wavelength NIR-fluorophore could be achieved by removing the electron-donating isopropyl group in **NIRQ_750_**, which would afford a hypochromic shift in absorbance. The maximum absorbance of **NIRQ_700_** was found to be blue-shifted, with a broad range in absorbance of 600–750 nm.

In 2017, Zhu et al. published an example of a fluorescent, NIR-emissive cyanostyryl azulene that was used for cell imaging [85]. Using cyanostyryl azulenes **19** and **20**, it was found that fluorescence was dependent on both the protonation state of the azulene and also the alkene geometry. Initial studies were performed on **(*Z*)-19** in CHCl_3_. Attempts at photoisomerisation to the corresponding (*E*)-alkene, using both UV and visible light, proved unsuccessful. Upon protonation of the azulene 5-membered ring, photoisomersation of [(*Z*)-19 + H]^+^ to [(*E*)-19 + H]^+^ was achieved by irradiation at 365 nm (Scheme 3). Both **(*Z*)-19** and [(*Z*)-19 + H]^+^ showed strong absorbances at 420 nm, corresponding to a S_0_→S_2_ transition. Whilst **(*Z*)-19** was found to be only very weakly fluorescent (λ_ex_ = 330 nm), [(*Z*)-19 + H]^+^ was found to be emissive at 500 nm at the same excitation wavelength. In contrast to the (*Z*)-isomers, the (*E*)-isomers showed a decrease in absorbance at 420 nm with a new, strong absorbance band at 680 nm suggesting an increase in favourability of the S_0_→S_1_ transition (and subsequent decrease in the S_0_→S_2_ transition). Fluorescence emission of [(*E*)-19 + H]^+^ was found to be greater than [(*Z*)-19 + H]^+^, but emission intensity of **(*E*)-19** (480 nm) was found to be the greatest of all. A new, weaker emission band at the NIR region of 700–900 nm was observed for **(*E*)-19**. B3LYP/6-31G studies of the photoisomerisation process suggested a twisted intramolecular charge transfer in the cyanostyryl group in the excited state of the *Z* isomer, affording a non-radiative decay pathway. The increased steric hindrance and rigidity of the *E* isomer are thought to disfavour the non-radiative charge transfer decay pathway, affording fluorescence emission.

By changing the nature of the phenolic tail group from an *n*-butyl chain to the polyethylene glycol chain PEG_2000_-yl, a water-soluble probe **(*Z*)-20** was generated. Identical results were obtained for **(*Z*)-20** in water for the photoisomerisation process. Similarly, dual emission bands were also found for ***(E*)-20**. However, unlike ***(E*)-19**, the quantum yield of the NIR emission of ***(E*)-20** (λ_em_~800 nm) was found to be roughly equal to that of the quantum yield of the visible emission band (λ_ex_ = 330 nm, λ_em_~400 nm) when excited at 710 nm. The intense NIR emission was attributed to the favourability of the S_0_→S_1_ transition allowing emission from the S_1_ excited stated, as well as the hindered non-radiative decay pathway in the (*E*)-isomer. The dual emission bands of ***(E*)-20** were exploited for cell imaging in MC3T3-E1 cell lines, which was contrasted with the **(*Z*)-20** isomer (Figure 17). Cell endocytosis was mediated by vesicle formation of the probes, as a result of the PEG tails. When excited at 405 or 640 nm, the (*E*)-isomer was clearly observed in the cell images. When contrasted to **(*Z*)-20**, some fluorescence was observed at the shortest excitation wavelength, whilst no fluorescence was seen at the longer wavelength. The group have further utilised similar photoswitching platforms for acidic sensing [86,87]. It should also be noted that changes in fluorescent emission of azulene derivatives upon photoswitching have previously been reported by Nica et al. [88].

Azulene fluorescent sensing for analyte detection is relatively under-developed. The azulene proton probe field has mainly seen use in the area of functional materials. The **Aal** field is rich with applications in peptide chemistry but this amino acid has not been used for analyte detection. Similarly, whilst cyanostyryl probe **20** was successfully used for cell imaging, no application in detection of a specific analyte was reported. To the best of our knowledge, our probe **AzuFluor^®^ 483-Bpin** is currently the only azulene probe that has been designed and employed for fluorescence bioimaging in response to specific analytes [89].

Chemodosimeter **AzuFluor^®^ 483-Bpin** was developed for the detection of reactive oxygen species (ROS) and reactive nitrogen species (RNS), exploiting the use of a boronate ester as a receptor. Upon reaction with ROS/RNS species, **AzuFluor^®^ 483-Bpin** was rapidly converted to 6-hydroxy species **21** (Scheme 4). The maximum absorption was observed at 327 nm for **AzuFluor^®^ 483-Bpin**, which was red-shifted to 350 nm in oxidized product **21**. **AzuFluor^®^ 483-Bpin** was found to be non-fluorescent, undergoing a fluorescent “turn-on” in the presence of ROS/RNS. Oxidized species **21** was found to emit at 483 nm (λ_ex_ = 350 nm), showing a broad Stokes shift of 133 nm.

DFT calculations (using the BP86 functional and 6-31G** basis set) showed a destabilization of the LUMO of **21** relative to **AzuFluor^®^ 483-Bpin**, affording a narrowing of the LUMO–LUMO+1 gap. The presence of the electron donating 6-hydroxy group increases the overall ICT character of **21**, leading to fluorescence emission.

The dosimeter was also evaluated for its *two-photon fluorescence* properties. Two-photon microscopy (TPM) [90] allows for excitation in the NIR region, enabling deep tissue penetration and greater localization of excitation [91]. As such, it has been exploited for a range of bioimaging applications [92]. Importantly the wavelengths required by TPM reduce the risk of autofluorescence from endogenous species. The maximum two-photon action cross-section of **AzuFluor^®^ 483-Bpin** upon addition of ROS was found to red-shift from 700 nm to 810 nm, affording a two-photon absorption cross-section of 320 GM for **21**. In vitro HeLa cell studies of **AzuFluor^®^ 483-Bpin** highlighted that fluorescence emission could be turned on in the presence of ONOO**^−^** to image cells. When excited at 800 nm, the emission intensity of **AzuFluor^®^ 483-Bpin** + ONOO**^−^** increased four-fold with respect to **AzuFluor^®^ 483-Bpin** (Figure 18). The probe was also shown to successfully undergo turn on fluorescence, and subsequent TPM imaging, when introduced to RAW 264.7 cells pretreated with inducers of endogenous ROS species. **AzuFluor^®^ 483-Bpin** was found to be non-cytotoxic and to be photostable for 1 h with and without the presence of ONOO**^−^**.

## 3. Conclusions

This review highlighted areas of the literature in which the azulene core has been employed as a fluorescent reporter moiety in the field of chemical probes. A summary of each system is reported in Table 1.

So far, the concept of azulene-based probes has been underutilised in the literature when compared with other common fluorophores (e.g., pyrene, coumarin), mainly being employed with regard to functional materials that are turn-on fluorescent protonation probes. One reason that may be hindering the growth of azulene sensors and dosimeters may be the relatively low quantum yield of azulene fluorophores (e.g., the quantum yield of **21** was determined to be 0.010). Therefore, the future of azulene probes may lie in increasing their emission efficiency. Various methods for achieving this have been reported for other classes of fluorophores, such as rigidification and the introduction of additional rings [93]; such approaches may prove to be equally applicable to azulenes.

The azulene core provides a highly attractive reporter motif when considering the creation of a novel probe species. As well as offering a fluorescence response, azulene is highly colourful. Importantly, the colour of azulene compounds can easily be predicted based on the location and electronic nature of its substituents [6]. Therefore, in comparison to other common fluorophores, azulene provides to the synthetic chemist a platform with which to create “designer probes”, whereby the fluorescence and colorimetric response can be fine-tuned. Once access to greater quantum-yielding azulene cores becomes available, greater application of azulene in the world of sensing will follow. Examples such as the fluoride probe by Eichen et al. and the ROS/RNS probe by Lewis et al. prove that such applications are possible. We believe that the future is bright for azulene.

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
