# Peer review of "Azulene—A Bright Core for Sensing and Imaging"

_molecules, 2021, doi:10.3390/molecules26020353_

Round 1
Reviewer 1 Report
Azulene is a hydrocarbon isomer of naphthalene known for its unusual colour and fluorescence properties. This review focused specifically on the diversity structures of Azulene derivatives. The review is subdivided into two sections. Section one discusses turn-on fluorescent sensors employing azulene, for which the literature is dominated with examples of the unusual phenomenon of azulene protonation-dependent fluorescence. Section two focusses on fluorescent azulenes that have been used in the context of biological sensing and imaging. In my opinion, this review can be published in Molecules after miner revision.
- In section one, several examples with picture of cell imaging should be added.
- For the molecules included in the manuscript, emphasized the azulene skeleton with color is better to know.
- In the title, the author indicated the sensing and imaging, so the sensing application for selective detection of analytes should be list as examples.
Author Response
Dear Reviewer,
Thank you for performing the necessary literature review process and assigning expert reviewers for our manuscript ‘Azulene – A Bright Core for Sensing and Imaging’, submitted for the Molecules special issue ‘Recent Advances in Molecular Sensors’ (Manuscript ID: molecules-1062713). We thank the reviewers for taking the time to evaluate our manuscript. The reviewers’ comments have been addressed and the following letter lists a point-by-point response to each comment (in each instance, the reviewer’s comments are in blue). The newer draft of the manuscript has been uploaded as an attachment.
Reviewer 1:
In section one, several examples with picture of cell imaging should be added.
- The referee makes an excellent suggestion that would increase the appeal of the review. We have added another appropriate image in Figure 17. We hope this review now includes all appropriate cell images from the literature.
For the molecules included in the manuscript, emphasized the azulene skeleton with color is better to know.
- Thank you for the suggestion. The azulene skeleton has been coloured blue throughout the review, making it clear to the reader how it has been incorporated.
In the title, the author indicated the sensing and imaging, so the sensing application for selective detection of analytes should be list as examples.
- A summary table has been added to the conclusion which lists each sensing/imaging application, which we think improves the overall review to make for easy access for the reader. This was a great suggestion, thanks.
Sincerely,
Lloyd Murfin
Reviewer 2 Report
The authors overview the performance of azulene as a fluorescent sensor (mainly for protons) and probes for bioimaging. To this aim, they revisit the most outstanding molecular structures reported in the bibliography in this biological context. The conducted bibliographic search seems to be exhaustive and cover most of the designed molecular structures to apply azulene for sensing and imaging. Besides, the results are briefly but clearly described to provide the reader an overall picture about the state of the art. Therefore, I think that this review could be suitable for publication in Molecules after some minor points are properly addressed. Such issues are related with the introduction and conclusion because I miss some additional information about the strengths and weaknesses of these dyes and an outlook. I will try to explain it in the following lines:
Introduction:
It seems that one of the main advantages of azulene is its tunable photophysical properties depending on the substitution pattern. What about the synthetic accessibility?? Is it straightforward and facile or tedious with many steps? Besides, it will be desirable to indetify which aspects maka especially appealing with regard to other fluorescent polihydrocarbons (anthracene, pirene, perylene) and dyes (coumarine, carbazole, fluorine, thiophene) working in the same spectral region. I mean briefly mention the main advantages of azulene with regard to other standard fluorophore in the UV-blue spectral window.
The last paragraph about dyes in the biological window is a little bit confusing because azulene absorbs and emits in the UV-blue region. It is true that along the discussion some examples are provided where azulene-based derivatives emits in the NIR (few times absorb in such region). But in such cases azulene is part of oligomers (as those based on fluorine) or replace part of well-known NIR dyes (squaranine or aminoacids). In the rest of sensors and probes based on azulene their spectral bands are placed in the blue-green part of the visible.
I think that in Figure 1 the LUMO+1 of naphthalene should be also added for comparison
Conclusions:
I think that in a review this section should be oriented like an outlook. Therefore, I recommend to rewrite this section and identify the weaknesses of these dyes and which properties should be improved. Indeed the authors mention that the future is bright for azulenes, but in which areas? What are actually the main research currents dealing with azulene? In the authors opinion what aspects should be improved to ameliorate its biophotonic performance??
Author Response
Dear Reviewer,
Thank you for performing the necessary literature review process and assigning expert reviewers for our manuscript ‘Azulene – A Bright Core for Sensing and Imaging’, submitted for the Molecules special issue ‘Recent Advances in Molecular Sensors’ (Manuscript ID: molecules-1062713). We thank the reviewers for taking the time to evaluate our manuscript. The reviewers’ comments have been addressed and the following letter lists a point-by-point response to each comment (in each instance, the reviewer’s comments are in blue). The newer draft of the manuscript has been uploaded as an attachment.
Reviewer 2:
It seems that one of the main advantages of azulene is its tunable photophysical properties depending on the substitution pattern. What about the synthetic accessibility?? Is it straightforward and facile or tedious with many steps?
- The reviewer raises an excellent point that would be of the utmost interest to the reader. An additional paragraph has been added with additional references that highlight common synthetic transformations of azulene.
Besides, it will be desirable to indetify which aspects maka especially appealing with regard to other fluorescent polihydrocarbons (anthracene, pirene, perylene) and dyes (coumarine, carbazole, fluorine, thiophene) working in the same spectral region. I mean briefly mention the main advantages of azulene with regard to other standard fluorophore in the UV-blue spectral window.
- An additional line has been added. This outlines that the nature of azulene is highly tuneable and predictable, both in terms of its fluorescence properties and colour. Therefore, azulene can be described as a ‘designer’ reporter moiety, and any fluorescent probe designed is also likely to act as a highly obvious colorimetric probe too. Based on the other points raised by the reviewer, this has been included in the ‘outlook’ section.
The last paragraph about dyes in the biological window is a little bit confusing because azulene absorbs and emits in the UV-blue region. It is true that along the discussion some examples are provided where azulene-based derivatives emits in the NIR (few times absorb in such region). But in such cases azulene is part of oligomers (as those based on fluorine) or replace part of well-known NIR dyes (squaranine or aminoacids). In the rest of sensors and probes based on azulene their spectral bands are placed in the blue-green part of the visible.
- We thank the reviewer for picking up this confusing section of text. The probe in question is a 2-photon probe, and as such is excited by NIR. An additional section of text has been added to clarify this key advantage, and we hope this is now clearer to the reader.
I think that in Figure 1 the LUMO+1 of naphthalene should be also added for comparison
- The naphthalene LUMO+1 has been added for comparison.
I think that in a review this section should be oriented like an outlook. Therefore, I recommend to rewrite this section and identify the weaknesses of these dyes and which properties should be improved. Indeed the authors mention that the future is bright for azulenes, but in which areas? What are actually the main research currents dealing with azulene? In the authors opinion what aspects should be improved to ameliorate its biophotonic performance??
- The conclusion has been rewritten in terms of an outlook. We have added in why we believe azulene to be a powerful sensing reporter, but also the disadvantages of it and where the future may lie in overcoming these.
Sincerely,
Lloyd Murfin